# Acalabrutinib May Offer a New Therapeutic Approach for Consolidation and Maintenance of Primary CNS Lymphoma with Expression of MYD88 and CD79B Gene Variants: A Case Report and Literature Review of Primary CNS Lymphoma in the BTKi Era

**DOI:** 10.3390/ijms262110521

**Published:** 2025-10-29

**Authors:** Eleanor Allison, Ashlea Campbell, Anne-Marie Watson, Brendan Beaton

**Affiliations:** 1Liverpool Hospital, Liverpool, NSW 2170, Australiaannemarie.watson@health.nsw.gov.au (A.-M.W.); 2New South Wales Health Pathology, Liverpool, NSW 2170, Australia; 3Liverpool Campus, University of New South Wales, Liverpool, NSW 2170, Australia; 4Campbelltown Campus, Western Sydney University, Campbelltown, NSW 2560, Australia

**Keywords:** primary CNS lymphoma, BTK inhibitor, MYD88 variant, CD79B variant

## Abstract

We present the case of a patient with primary CNS lymphoma (PCNSL), with MYD88 and CD79B gene variants, who was unable to complete standard induction and consolidation treatment due to toxicity and co-morbidities after three cycles of MATRix. Although he had responded to truncated induction, acalabrutinib, the BTK inhibitor, was used in an attempt to consolidate and maintain his response. He has an ongoing remission at 18 months of follow-up. Following the case presentation, we provide a review of PCNSL, the evolution of therapy, and how BTK inhibitors are now emerging treatments incorporated into the salvage of relapsed and refractory disease and into first-line treatment in some clinical trials. This is the first reported case in the literature of acalabrutinib use for consolidation and maintenance of PCNSL. We hope this can support clinical trial design for BTKi use in this setting in the future.

## 1. Introduction

Primary central nervous system lymphoma (PCNSL) is a rare, aggressive extranodal B-cell malignancy confined to the CNS. Standard therapy combines high-dose methotrexate induction with consolidation utilising autologous stem cell transplant or whole-brain radiotherapy, but many patients—particularly older adults or those with comorbidities or treatment-related toxicity—are ineligible for these approaches, often resulting in poorer outcomes.

Molecular profiling has revealed frequent MYD88 and CD79B mutations, characteristic of the MCD subtype, in PCNSL. These alterations drive constitutive B-cell receptor signalling via *Bruton’s tyrosine kinase* (BTK), providing a biologically rational target for therapeutic intervention. BTK inhibitors (BTKi) penetrate the blood–brain barrier and show efficacy in relapsed or refractory PCNSL, yet evidence for their use as upfront consolidation or maintenance remains limited.

Here, we present a case of a patient with PCNSL who was unable to tolerate standard consolidation due to severe treatment-related toxicity and comorbidities. The patient received acalabrutinib as a consolidation and maintenance strategy, achieving sustained remission with functional recovery. This case highlights a critical knowledge gap regarding the role of BTK inhibition in patients unsuitable for standard therapy and underscores the need for systematic evaluation of BTKi as consolidation or maintenance in PCNSL.

## 2. Case

A 58-year-old male with a history of obesity, poorly controlled type 2 diabetes, chronic hazardous alcohol use, and a 50 pack-year cigarette smoking history presented to the emergency department with disorientation, disordered speech and thought content, visual disturbance, and chest pain.

Computerised tomography (CT) of the brain showed several lesions, most prominent in the right parieto-occipital area. Subsequent magnetic resonance imaging (MRI) of the brain demonstrated cerebral lesions with diffusion weighted imaging (DWI) and T2 fluid-attenuated inversion recovery (FLAIR) enhancement of right parieto-occipital lesions (Figure 1A), favouring a diagnosis of cerebral lymphoma.

Cerebro-spinal fluid (CSF) examination revealed no atypical lymphocytes or other malignant cells on cyto-spin and no clonal B lymphocytes on flow cytometry immunophenotyping studies. Total protein was elevated at 2.24 g/L (RR: 0.15–0.45 g/L); glucose was elevated at 5.6 mmol/L (RR: 2.5–4.5 mmol/L); and serum angiotensin-converting enzyme level was normal at <8 U/L. Assessment for mycobacterium tuberculosis (TB) was negative by polymerase chain reaction (PCR) for TB DNA and negative by culture after 8 weeks; fungal culture was also negative. CSF nucleic acid PCR did not detect Enterovirus, Herpes virus types 1 and 2, Neisseria meningitidis, Strep pneumoniae, Varicella zoster, or Parechovirus.

A positron emission tomography (PET) scan showed no fluorodeoxyglucose (FDG)-avid lesions outside the CNS, with marked FDG avidity of the right parieto-occipital lesions (Figure 1B). There was no ocular involvement with lymphoma. MRI of the spine was not performed. Testicular ultrasound showed no lesions suspicious for testicular lymphoma.

A stereotactic brain biopsy was performed, which demonstrated sheets of large cells with immunohistochemistry positive for CD20, CD79a, CD45, BCL-6, PAX5, and MUM1. CD10 showed weak cytoplasmic staining only and was considered negative. BCL-2 was negative, and Ki67 was 90% (Figure 2). EBER-ISH was negative. MYC was not rearranged on FISH analysis. A diagnosis of primary CNS diffuse large B-cell lymphoma (PCNSL) was confirmed. Molecular testing revealed variants in the following genes: *MYD88* (c.794 T>C; p.Leu265Pro) with variant read fraction (VRF) of 13%; *CD79B* (c.587 A>C; p.Tyr196Ser) with a VRF of 22%; and *CARD11* (c.1135 C>T; p.Arg379Trp) with a VRF of 9%. (Testing was performed on the Illumina Novoseq X plus platform, and patient consent was obtained).

Prior to treatment, neuropsychological testing revealed significant impairment in executive function. Baseline organ function was satisfactory, with an estimated glomerular filtration rate (eGFR) of >90 mL/min, normal liver function tests, and normal left ventricular and systolic function with an ejection fraction of approximately 60%. Baseline lung function was not performed. Plain chest X-ray was unremarkable. The Charlson Comorbidity Index score was 7. The Karnofsky Performance Scale score was 25. The Eastern Cooperative Oncology Group (ECOG) score was 2.

Following discussion at the institutional lymphoma multidisciplinary team (MDT) meeting, the patient commenced the MATRix protocol (methotrexate, Ara-C, thiotepa, rituximab), planning for four cycles of induction followed by carmustine/thiotepa high-dose conditioning therapy and autologous stem cell transplant (ASCT). MRI after two cycles showed marked reduction in the right parieto-occipital lesion. However, following the third MATRix cycle, treatment was complicated by severe methotrexate-induced renal impairment with delayed methotrexate clearance, biopsy-proven cryptogenic organising pneumonia, panhypopituitarism, and COVID-19 infection. Post-cycle 3 MRI (Figure 1C) and cerebral PET (Figure 1D) showed further lesion reduction and decreased FDG uptake (Deauville 3), consistent with complete metabolic response. Given the patient’s intolerance to intensive chemotherapy, poor performance status with no improvement with therapy, and contraindications to both whole-brain radiotherapy and autologous transplantation, a *Bruton tyrosine kinase* (BTK) inhibitor was selected as a safer, targeted consolidation approach with favourable tolerability and central nervous system penetration. Given the patient’s significant cardiovascular risk factors, the second-generation BTK inhibitor acalabrutinib was selected for its improved safety profile. As BTKi use in PCNSL is not reimbursed under the Australian Pharmaceutical Benefits Scheme, compassionate access was obtained from AstraZeneca (Macquarie Park, NSW, Australia)). Treatment commenced at a standard dose of 100 mg twice daily.

The patient tolerated acalabrutinib without toxicity and remains in clinical and radiological remission (18 months at the time of finalising this manuscript) after treatment initiation. Cryptogenic organising pneumonia was managed with a three-month tapering course of prednisolone, resulting in complete clinical and radiological resolution. Physical function improved with rehabilitation, and cognitive and executive capacity gradually improved, though no formal neuropsychological testing was repeated. He was discharged from the hospital into a supported living facility, where he lives mostly independently. At 18 months after initiating acalabrutinib, the patient’s Karnofsky Performance Status score was 65, and the ECOG score was 1. Despite improvements in physical and cognitive function, autologous stem cell transplantation was deemed high-risk with limited evidence for benefit more than 12 months post-induction. Given that he maintains remission while taking acalabrutinib with no toxicity, and in the absence of evidence supporting cessation of acalabrutinib in this setting, the risk of relapse upon discontinuation remains a significant concern. The ongoing use of acalabrutinib will be re-discussed in relation to the patient’s progress moving forward and as further evidence of BTKi use in this setting continues to emerge.

## 3. Outline of Primary CNS Lymphoma

Primary central nervous system lymphoma (PCNSL) is an extra-nodal non-Hodgkin lymphoma occurring exclusively within the CNS, including the brain parenchyma, leptomeninges, spinal cord, cranial nerves, or eyes [1]. The majority are histologically classified as diffuse large B-cell (DLBCL) and are aggressive tumours with generally poor prognosis [2]. PCNSL is rare and accounts for only 4–6% of all extra-nodal lymphomas, with a median age at diagnosis of 68 years [3].

According to the 2022 WHO classification of lymphoid malignancies, PCNSL sits under the umbrella of primary large B-cell lymphomas of immune-privileged sites, since they share common biological, immunophenotypic, and molecular features with lymphomas arising in immune sanctuary sites such as testicular and vitreoretinal lymphoma [2]. A subset of PCNSL is associated with immunosuppression and/or HIV infection, the majority of which are positive for lymphotropic Epstein–Barr Virus (EBV). These EBV-associated PCNSL are clinically, biologically, and molecularly distinct from those arising de novo [1], and as such, treatment paradigms diverge significantly.

The clinical symptoms of PCNSL are dictated by the location of lesions. Sixty-five percent of patients have a solitary lesion [1]. Parenchymal involvement is present in most cases and classically presents with focal neurological deficits such as weakness, sensory change, visual disturbance, neuropsychiatric and behavioural changes, and symptoms of increased intracranial pressure such as headache or seizures [1]. Isolated leptomeningeal or spinal cord disease is rare, and symptoms typically overlap with those of parenchymal disease. Vitreoretinal lymphoma (VRL) usually occurs concomitant to parenchymal lesions and may be present even in the absence of visual disturbance [1]. B-symptoms such as night sweats, weight loss, and fever are rare.

Time from onset of symptoms to diagnosis is typically rapid after the development of sudden focal neurological symptoms but may be delayed in those presenting with insidious neurological decline. The gold standard for diagnostic imaging is gadolinium-enhanced MRI of the brain and spinal cord, demonstrating T2 hypointense lesions, with surrounding vasogenic oedema and homogeneous, diffuse contrast enhancement. Stereotactic biopsy of a lesion is highly desirable to confirm the diagnosis [3,4]. Those with isolated vitreoretinal involvement should undergo vitrectomy. If it is safe, a diagnostic lumbar puncture should be performed, which may help with diagnosis if tissue is not available. CSF investigations often demonstrate elevated protein levels, reflecting a disrupted blood–brain barrier (BBB), normal glucose, and elevated lymphocyte count, which may show clonality by flow cytometric lymphoid immunophenotyping. Raised CSF protein is a feature of poorer prognosis. FDG-PET scan is performed to complete conventional lymphoma staging and rule out systemic lymphoma. Additional investigations with bone marrow biopsy and testicular ultrasound should always be considered to exclude lymphoma involvement.

Treatment selection must involve an evaluation of medical comorbidities and functional status. Age and performance status are major predictors of outcome and will dictate the appropriate tolerable therapeutic options [5,6]. There are many tools to assist functional assessment, including the Comprehensive Geriatric Assessment (CGA), the Charlson Comorbidity Index (CCI), the Cumulative Illness Rating Scale for Geriatrics (CIRS-G), and the comparatively shorter G8. Though widely advocated, holistic and comprehensive geriatric assessments are cumbersome and time-consuming to complete, translating to poor adherence in practice [6]. Knowledge of function immediately prior to the onset of neurological symptoms is important, as more intensive therapy regimens may be avoided if patients are thought to be too frail or whose ECOG is high, unless this is disease-related and pre-morbid baseline function was good. Function and ECOG may improve with commencement of treatment. Dynamic decisions need to be made based on changes in function, ECOG, and mentation. It is important to embed functional assessment into clinical practice for patients with CNS lymphoma and include repeat assessments over the course of therapy, at completion, and into remission.

## 4. Molecular Characterisation

Considerable advances in understanding the unique biology of PCNSL have been made in recent years. Tumour cells are invariably positive for CD20, CD79a, and PAX5, and the vast majority are of the activated B-cell (ABC) subtype (by cell of origin), CD10-negative, and BCL6- and MUM1-positive [1,7]. The pathogenesis of PCNSL lies in the dysregulation of key cell signalling pathways that ultimately lead to NF-kB activation and B-cell proliferation and survival [8]. There are typical recurrent mutations primarily affecting MYD88^L265P^ and CD79B, placing PCNSL in alignment with the MCD (**M**YD88/**CD**79B-mutated) lymphoma molecular subtype of DLBCL which is also characterized by poor prognosis and extra-nodal involvement, especially in immune-privileged sites [8,9]. Of significance, lymphoproliferative disorders with variants in these genes have been proven to be highly sensitive to the use of BTK inhibition [10].

Genetic inactivation of MHC class I and II, and β2-microglobulin, with subsequent loss of protein expression, together with a 9p24 gain, results in overexpression of PD-L1/2 in 30–50% of cases, enabling immune escape [2,8,11,12]. Furthermore, PCNSLs exhibit significantly more focal deletions of HLA-D (6p21), which may contribute to immune evasion by reducing immune surveillance [13]. In addition, aberrant somatic hypermutation and kataegis (mutational hotspots) play a significant role in shaping the mutational landscape of PCNSL [13].

Emerging evidence has shown that PCNSLs can be differentiated from systemic DLBCLs using RNA sequencing that demonstrates distinct transcriptomic signatures and higher TERT gene expression [13]. This may be utilized in cases where differentiating PCNSL from DLBCL with systemic involvement is difficult.

There are currently no established subtypes of PCNSL, and when taken altogether, there is significant heterogeneity in the molecular signature. However, when grouped with respect to genomic characteristics (whole-exome sequencing, RNA-seq, copy number alterations, DNA methylation) and clinico-radiological data, there may emerge certain clusters with prognostic significance [14,15]. Hernández-Verdin et al. have used RNA sequencing and proteomics to propose clustering of mutational signatures together, and hypothesise that some clusters are more sensitive to Bruton’s tyrosine kinase (BTK) inhibitors, while others have high JAK-STAT activity and mutated SOCS1 (JAK1 repressor), making them potentially responsive to JAK1 inhibitors [15]. These distinctive clusters are further highlighted when comparing EBV status. Hai et al. have demonstrated that EBV+ and EBV- PCNSL harbour distinct transcriptional and epigenetic profiles, specifically that BCR and BTK signalling is particularly active in EBV negative PCNSL [16]. The reliance on these pathways makes EBV-negative PCNSL suitable for targeting with BTKis. Further research is required to validate these potential subtypes and translate them for clinical use.

## 5. Non-Invasive Diagnostics and Monitoring

While stereotactic brain biopsy is the diagnostic gold standard, this can be difficult to achieve safely given lesions are sometimes in anatomically challenging and high-risk locations. If a patient is commenced on corticosteroids prior to biopsy, this may obscure histological diagnosis by causing necrosis of the tumour. Traditional methods of CSF evaluation with cytology are often only positive in cases involving the leptomeninges or parts of the brain with intraventricular extension [17]. Therefore, there remains a need to develop non-invasive techniques to diagnose PCNSL that could either complement or, in certain circumstances, replace the need for a brain biopsy.

Identification of the MYD88^L265P^ mutation and elevated levels of IL-10 (>2 pg/mL) in CSF or vitreous fluid may be diagnostic for those unfit to undergo a brain biopsy, particularly where samples are pauci-particulate [8,18,19]. CSF proteomics involves large-scale protein profiling using mass spectrometry or multiplex assays to identify and quantify proteins reflective of CNS pathology. Several candidate biomarkers—most notably interleukin-10 (IL-10), the IL-10/IL-6 ratio, CXCL13, and free light chains—demonstrate diagnostic potential, while more recent mass spectrometry studies have identified additional proteins such as LCP1, SGCE, and AGRN that correlate with treatment response and tumour burden [20].

Elevated IL-10 at diagnosis has been shown to correlate with worse overall survival, and persistent elevation at the end of treatment may predict rapid relapse [19,21]. Other proposed biomarkers include CSF CXCL13 (a B lymphocyte chemo-attractant that binds to the CXCR5 receptor) and neopterin (a nonspecific marker of the type 1 T-helper cell–related cellular immune response), which have shown sensitivity in the diagnosis of cerebral lymphoma. However, they lack specificity and may not differentiate PCNSL from SCNSL [19]. Extrapolating from their emerging utility in other non-Hodgkin lymphomas, soluble CD19, soluble CD27, and immunoglobulin free light chains may also have merit in contributing to an overall diagnostic picture, but require further research [19].

Currently, CSF proteomics remains primarily a research tool, with clinical application largely limited to cytokine-based assays such as IL-10 quantification. However, emerging evidence supports its potential integration into clinical workflows to aid diagnosis, risk stratification, and real-time monitoring of treatment efficacy or relapse [20,21]. Multi-omic approaches combining CSF proteomics with circulating tumour DNA or methylation profiling have achieved high diagnostic discrimination between PCNSL and other CNS malignancies, suggesting that proteomic signatures could eventually reduce reliance on invasive brain biopsy [22]. Future advances in assay standardisation, validation in large cohorts, and integration into diagnostic algorithms are expected to accelerate the transition of CSF proteomics from discovery science to a clinically actionable tool in PCNSL management.

Detection of *MYD88* mutations, particularly the L265P hotspot, is most commonly performed using allele-specific PCR (AS-PCR) or digital droplet PCR (ddPCR), which offer high analytical sensitivity and rapid turnaround. AS-PCR is widely used in diagnostic laboratories due to its simplicity, low cost, and ability to detect mutant allele frequencies as low as 1–5%. ddPCR provides even greater sensitivity, detecting variants at frequencies below 1%, and allows quantitative assessment of mutant allele burden, making it useful in samples with low tumour content [23]. Next-generation sequencing (NGS) is increasingly incorporated into broader molecular panels to detect both L265P and non-canonical *MYD88* variants, though it is less sensitive and more resource-intensive than targeted PCR-based methods.

Use of cell-free DNA (cfDNA) and the proportion of tumour-derived cfDNA (ctDNA) represents further opportunities for non-invasive diagnosis and monitoring of malignancies. Targeted sequencing of ctDNA using a variety of highly sensitive methods, including quantitative PCR, TaqMan PCR, next-generation sequencing, and ddPCR, can identify somatic mutations and gene alterations that can then serve as a molecular marker of disease status [19,24].

The particular challenge of PCNSL lies in the relatively intact blood–brain barrier, which limits the release of ctDNA into the bloodstream but concentrates it relatively in the CSF. In a small study of six cases of CNS-restricted lymphoma, detection of the MYD88^L265P^ mutation in CSF ctDNA was more sensitive at detecting disease than traditional methods, predicting relapse even while CSF flow cytometry and cytology remained negative [25]. Moreover, as a consequence of disease and treatment, the blood–brain barrier in PCNSL is impaired, allowing detection of ctDNA in the plasma and urine, though at reduced rates compared with CSF [25,26,27,28]. CtDNA could also serve as a biomarker for risk stratification and prediction of outcome, as concentrations correlate with total radiographic tumour volumes measured by MRI, and higher ctDNA concentrations in any fluid at the time of diagnosis along with its persistence during treatment, correlates with poorer prognosis [29,30]. The diagnostic yield is often reduced by low tumour burden and limited DNA quantity in CSF, which can lead to false-negative results. Furthermore, *MYD88* mutations are not universally present in CNS lymphomas, and low-level detection in the absence of supportive cytological or radiological findings may be difficult to interpret. From a technical perspective, DNA degradation, contamination risk, and variability in sample processing can compromise assay performance, while conventional sequencing lacks the sensitivity required for CSF analysis [31,32]. Highly sensitive methods such as AS-PCR or ddPCR are preferred, but they are typically restricted to detection of the common L265P hotspot mutation. Consequently, *MYD88* testing in CSF should be interpreted cautiously and integrated with clinical and radiological context to ensure diagnostic accuracy.

Response to treatment is currently based on serial MRI monitoring. However, there can be difficulty differentiating active disease from disease-free but damaged tissue when assessing small volumes of residual tumour. Identifying the molecular signature at diagnosis and subsequent measurement of ctDNA may represent further opportunities for accurate monitoring of response to treatment and detection of molecular relapse, even in radiologic remission (or if there is uncertainty based on nonspecific image findings) [25]. As in other lymphomas, further studies and validation are required before its translation into risk stratification and clinical management.

Identifying PCNSL as being of the MCD subtype, either by identifying a MYD88^L265P^ mutation or a variant of CD79B, may have potential therapeutic implications, as tumours of this subtype are sensitive to treatment using BTK inhibitors.

## 6. Treatment Evolution

### 6.1. Induction

Historically, corticosteroids and whole-brain radiotherapy were the only treatments available for PCNSL. This translated to dismal survival outcomes, with overall 5-year survival of <5% [33]. The introduction of high-dose methotrexate (HD-MTX) at doses achieving cytotoxic CSF concentrations (>3 g/m^2^) [34], followed by the addition of high-dose cytarabine, improved outcomes [35]. The addition of rituximab, the anti-CD20 monoclonal antibody, to chemotherapy combinations further improved outcomes [36]. The landmark IELSG32 trial [37] proved the intensive four-drug combination induction therapy with MATRix (HD-MTX, cytarabine, thiotepa, and rituximab) performed superiorly to two-drug (HD-MTX/cytarabine) or three-drug (rituximab/HD-MTX/cytarabine [28]) combinations. Seven-year follow-up demonstrated overall survival of 56%, rising to 70% for those who had MATRix followed by consolidation [38]. A real-world, multi-centre retrospective analysis of 156 patients with newly diagnosed PCNSL treated with MATRix, recapitulated these findings, reporting a two-year progression-free survival (PFS) of 56% and overall survival (OS) of 64.1% [39]. The role of routine intrathecal chemotherapy has not been established and is not recommended [3]. In patients deemed ineligible for high-dose methotrexate treatment but suitable for treatment with palliative intent, oral temozolomide and/or whole-brain radiotherapy and/or best supportive care remain options [40,41].

### 6.2. Consolidation

For fit patients who do not progress (i.e., in CR or PR) while on induction therapy, standard of care remains thiotepa-based high-dose conditioning chemotherapy and ASCT. If a patient is not considered eligible for transplant, whole-brain radiotherapy (WBRT) can be considered. However, this comes with clinically significant toxicities, including impairment in attention, executive function, and memory [42]. A comparison of outcomes between WBRT and ASCT has been assessed in two randomised studies, with Houllier et al. [43] demonstrating poorer long-term survival in the WBRT group, but Ferreri et al. [44] suggested equipoise. Other options for those not eligible for transplant include maintenance with alkylating agents such as temozolomide or procarbazine.

### 6.3. Relapsed/Refractory Disease

Up to 25% of patients treated with the best available therapy experience refractory disease, and a further 25% experience relapse after initial response. The outcomes for these patients are extremely poor [3]. Salvage options include HD-cytarabine or HD-ifosfamide-based chemotherapy followed by consolidative ASCT. Where available, patients should be enrolled in clinical trials.

### 6.4. Treatment in the Elderly and Frail

Since its publication in 2016 [37], MATRix has been adopted as first-line therapy for fit patients under the age of 70. However, with the median age of presentation in the seventh decade [5], it is common for patients to need dose or treatment modifications. The diagnosis in elderly patients is more likely to be delayed because subtle symptoms or mild cognitive decline are dismissed as age-related changes [42] or can be attributed to other health issues. This not only corresponds to poorer baseline functional status at diagnosis, but delayed diagnosis is an independent predictor of inferior survival [42].

There is strong evidence that outcomes are linked to the dose of methotrexate, with those treated with lower doses having considerably worse outcomes [39]. An analysis of outcomes in elderly patients treated with methotrexate-based regimens found that the higher relative intensity of methotrexate dosing received resulted in better outcomes [45]. Options for lower-intensity therapy include the R-MP protocol (rituximab, high-dose methotrexate, and procarbazine induction, followed by 4-weekly maintenance treatment with procarbazine), which demonstrated a 2-year progression-free survival of 34.9% in the PRIMAIN study [46].

Treatment challenges in the elderly relate to balancing toxicity while maximising chemotherapy dose to ensure cytotoxic concentrations in the CNS. Issues include intolerance to the large volume of adjunct bicarbonate fluids required to facilitate renal excretion of high-dose methotrexate, resulting in fluid overload and potentiating heart failure. Conversely, poorer baseline renal function makes patients more vulnerable to renal toxicity. Elderly patients are more vulnerable to the side effects of glucocorticoids, including cognitive and behavioural disturbance, as well as hyperglycaemia [42]. In a retrospective review of 192 elderly patients (aged over 65 years) treated with a methotrexate-based regimen, treatment-related mortality was 6.8%, which, while comparable to the 6% in the IELSG32 trial, meant that around one-third of patients had significant dose modifications [45].

While treatment for CNS lymphoma can be challenging, and the outcomes suboptimal, recent trials may provide more tolerable therapy, with the hope of better outcomes. The single-arm, phase 2 MARTA trial enrolled 51 patients, 53% of whom had an ECOG of 2 or more and a median age of 71 years and administered a modified MATRix regimen by omitting thiotepa during a shortened induction, followed by a modified busulfan and thiotepa conditioning for ASCT. While it did not meet its primary efficacy endpoint, it demonstrated a 1-year overall survival of 80.6% in those completing treatment, which is comparable to younger cohorts [47]. This challenges the traditional wisdom of avoiding transplant in older patients by minimising upfront toxicity and may serve as a backbone for future research. Furthermore, current research is underway, investigating the efficacy of emerging therapies.

### 6.5. Emerging Novel Therapies

Given both the limitations and toxicities of current therapies, particularly in the elderly and in relapse, there has been much interest in novel therapies that may complement or even replace conventional chemotherapy. Among the most promising and well-studied therapies are the BTKis. However, there are a number of other clinical developments, including treatment with immune modulators such as lenalidomide, immune checkpoint inhibitors, bispecific antibodies and chimeric antigen receptor T-cells (CAR-T).

### 6.6. Bruton’s Tyrosine Kinase, B-Cell Activation, and BTK Inhibitors

Following antigen binding to the extracellular motif of the B-cell receptor, a chain reaction of intracellular signalling commences, including phosphorylation of immunoreceptor tyrosine-based activation motifs, recruitment of SYK tyrosine kinase, and phosphorylation of the BLNK adaptor protein, which then recruits and activates BTK. BTK is a cytoplasmic tyrosine kinase that, when phosphorylated following this initial chain reaction, goes on downstream to activate phospholipase Cy2 and, subsequently, inositol triphosphate and diacylglycerol. These are both important second messengers in NF-kB transcription factor upregulation, leading to B-cell proliferation, differentiation, and antibody production [48]. Small molecules that target and inhibit BTK, and hence downstream B-cell activation and proliferation, have emerged as highly effective for treating B-cell malignancies and autoimmune diseases driven by dysfunction in B-cells [10,49].

It has been recognised that variants in MYD88 and CD79B can predict sensitivity to BTK inhibition in B-cell lymphoproliferative disorders [50]. MYD88 is an adaptor protein that can be activated downstream of Toll-like receptors and IL1R family members, which activate B-cells and play an important part in innate immunity. Variants in MYD88, particularly the hotspot L265P variant, are commonly found in B-cell lymphoproliferative diseases, and are thought to constitutively activate NF-kB and hence induce B-cell activation [51]. CD79B is a subunit of the B-cell receptor, forming a heterodimer with CD79A, and is crucial for B-cell activation. Mutations in CD79B are known to lead to constitutive B-cell activation via activation of SYK, BTK, Pi3K, and eventually NF-kB, and hence lymphoma development through uncontrolled B-cell proliferation and survival [52]. BTK is a common kinase to both MYD88 and CD79B activation pathways; therefore, in lymphomas that are driven by constitutive activation of B-cell survival, it can be exploited by using inhibitors to prevent unregulated B-cell growth [50].

BTK inhibitors are well established as treatments for chronic lymphocytic leukaemia, mantle cell lymphoma, and Waldenström’s macroglobulinaemia. Subsequently, interest has turned toward use in other types of lymphoproliferative disease. BTK inhibitors can penetrate the blood–brain barrier and block the chronic B-cell receptor signalling that occurs in PCNSL, which frequently has mutations of MYD88 and/or CD79B, thereby inducing apoptosis of these lymphoma cells. There is a rational place for BTK inhibition in PCNSL [53,54].

Ibrutinib was the first-in-class covalent BTK inhibitor, binding to Cys-481, the active site of BTK. It has less specificity than BTK compared with later generations, demonstrating binding to cysteine residues in other kinases, including epidermal growth factor receptor family kinases, SRC kinases, and TEC kinases. These off-target bindings may cause side effects such as diarrhoea, dermatitis, atrial fibrillation, hypertension, and increased bleeding risk. Second-generation BTKis include acalabrutinib, zanubrutinib, tirabrutinib, and orelabrutinib. They exhibit higher target specificity than ibrutinib but still cause some noteworthy adverse effects, including headache and cough with acalabrutinib, and neutropenia with zanubrutinib [10].

BTKis are small molecules and have been proven to cross the intact blood–brain barrier, with ibrutinib, acalabrutinib, zanubruitinb, and tirabrutinib detectable in the CSF at concentrations associated with anti-tumour activity between 2 and 24 h after an oral dose [55,56,57]. In animal models, ibrutinib and tirabrutinib have demonstrated higher and more sustained CSF concentrations than zanabrutinib [58]. However, this has been refuted in humans following a case series of 13 patients with CNS involvement of DLBCL, which reported excellent BBB penetration of zanubrutinib [59]. Furthermore, rodent models of gliomas demonstrate that ibrutinib may also increase BBB permeability by causing junctional disruption and impairing efflux ports [60]. This enables increased concentrations of chemotherapies at the tumour site while limiting systemic toxicity, lending strength to the use of BTKis as upfront, adjunct therapy.

There have been two prospective studies evaluating ibrutinib monotherapy. In 2017, there was a phase 1/2 study of ibrutinib monotherapy in 13 patients with R/R PCNSL, achieving a response rate of 77% and a complete response rate of 38% [61]. Recent publication by the same group, with additional patient enrolment and 3-year follow-up, showed a response in 23/31 patients with PCNSL, with a median PFS of 4.5 months (95%CI: 2.8–9.2) and 1-year PFS of 23.7% (95%CI: 12.4–45.1%) [62]. The iLOC study was a phase 2 study of 44 patients with relapsed or refractory large B-cell primary oculo-cerebral lymphoma treated with ibrutinib monotherapy [63]. Short-term overall response rate after 2 months of treatment was 59%, including 10 complete responses (19%). Longer-term follow-up (median 66 months) showed a median overall survival of 20 months (IQR: 7–29 months) and PFS of 4.8 months (IQR: 3–13 months) [64].

Ibrutinib has also been studied in combination with chemoimmunotherapy (DA-TEDDi-R) (rituximab, liposomal doxorubicin, temozolomide, etoposide, and dexamethasone). In 14 evaluable patients (median age 66 years of an initial cohort of 18 patients) with R/R PCNSL, ORR was 91%, with 86% achieving CR/CRu. Of these, 57% had PFS at a median of 15.5 months of follow-up (range 8–27 months). Median PFS was calculated at 15.3 months. With limited follow-up at the time of reporting, median OS was not reached, with 51.3% alive at 1 year. Increased aspergillus infection was noted in this study [54].

Tirabrutinib is approved for use for R/R PCNSL in Japan, South Korea, and Taiwan. Three-year follow-up of 44 patients with R/R PCNSL treated with tirabrutinib monotherapy demonstrated an overall response rate of 63.6% (95% CI: 47.8–77.6) and median overall survival not reached [65,66]. The PROSPECT study (NCT04947319) has just presented its phase 2 data using tirabrutinib monotherapy in R/R PCNSL. Forty-eight patients (median age 65.6 years), with a median follow-up 11.2 months, showed an ORR of 66.7%, with a CR/CRu of 43.8%. Median DOR was 9.3 months. TRAEs of grade 3 or greater were experienced in 27.1% of patients [67].

Zanubrutinib has been used in combination with immunochemotherapy, with a case report of good response from the addition of zanubrutinib to HD-MTX and rituximab, which had been refractory after two cycles, allowing the patient to proceed to ASCT, with survival of at least 11 months after diagnosis [68]. In a retrospective analysis of the up-front use of BTKi in newly diagnosed PCNSL, 19 patients (median age 57) were treated with zanubrutinib and HD-MTX. The median follow-up duration was 14.7 months, with an ORR of 84.2%, and an OS rate of 94.1% [69]. ctDNA was assessed in CSF throughout treatment and correlated with radiologic response and predicted progressive disease while on maintenance treatment. The selection criteria for the use of zanubrutinib were not clear. Although a small study, this robust survival rate merits further investigation with a prospective study. The PRiZM+ study (ISRCTN90634455) has just presented its phase 2 data using zanubrutinib monotherapy in R/R PCNSL. Of 20 evaluable patients (median age 64 years), ORR was 55%, and CR/CRu was 35%, as determined by central neuroradiological review. With a median follow-up of 19 months, median PFS was 53%, and OS was 37%. Ten SAEs were reported [70].

Acalabrutinib is a second-generation BTKi currently being prospectively evaluated in clinical trials investigating its efficacy in R/R PCNSL as monotherapy (NCT04548648, NCT04906902) or in combination with the PD-1 inhibitor durvalumab (NCT04462328); these trials are ongoing. The latter trial reported phase 1 data on ten patients (median of two prior lines of therapy) with a 40% CR/CRu. Importantly, this communication had a correlative science arm testing ACP-196 (acalabrutinib) and ACP-5862 (the active metabolite of acalabrutinib) levels in plasma and CSF. ACP-196 was detected in CSF with a median concentration 4.6 ng/dL (range 2.75–10.6 ng/dL), and ACP-5862 had median concentration of 7.06 ng/dL, indicating penetrance of acalabrutinib and its active metabolite into the patients’ CSF [71]. Though no case reports have yet been published for the use of acalabrutinib monotherapy in PCNSL, there are reports of its effective use in combination with immunochemotherapy and radiotherapy for EBV-positive post-transplant lymphoproliferative disease following allogeneic stem cell transplant treating acute myeloid leukaemia [72]; monotherapy leading to CR of mantle cell lymphoma with CNS involvement [73]; and durable remission with acalabrutinib monotherapy in a patient with leptomeningeal disease secondary to relapsed mantle cell lymphoma [74].

A retrospective review of 82 patients with newly diagnosed PCNSL treated across three Chinese hospitals found that the 26 patients who received BTKi combined with chemotherapy had statistically significantly improved outcomes compared with 49 patients who had chemotherapy alone. At median follow up of 28 months, the median overall survival of the BTKi group was not reached, compared with the chemotherapy group at 47.8 months (IQR: 32.5–63.1 months) (*p* = 0.038) [75]. Notwithstanding the limitations of the retrospective, uncontrolled nature of this review, it adds strength to the use of upfront BTKis.

Another recent systematic review and meta-analysis of the safety and efficacy of BTKis in all types of CNS lymphoma (primary and secondary) included 21 studies involving 368 patients [76]. There were seven studies (35 patients) that used BTKis in the upfront setting. However, due to the small sample size, no pooled analysis was undertaken. In the R/R setting, 10 studies (166 patients) using monotherapy showed a pooled overall response rate of 60% (95% CI: 50–71%), and in 12 studies (176 patients) using combination therapy, the ORR was 78% (95% CI: 68–86%). With regard to the significance of MYD88 or CD79B mutational status, a review found that the effect of BTK inhibitors was independent of MYD88 or CD79B mutational status [77].

As the molecular landscape of PCNSL is further elucidated, there is a growing understanding of its effect on the variable response to BTKi. Mutations in caspase recruitment domain family member 11 (CARD11) are found in 11–30% of PCNSL and can promote resistance to single-agent ibrutinib [78]. CD79B mutations may upregulate PI3K/mTOR signalling to convey resistance [79]. Comprehensive genomic profiling is increasingly being used to identify biomarkers that may be predictive of immunotherapy efficacy. However, findings have been conflicting, with some proposing that high tumour mutational burden is predictive of a good response to immunotherapy [80], whereas others find patients with complex genomic features respond poorly to ibrutinib [81].

### 6.7. Other Therapies

CAR-T therapy is an established treatment in relapsed and refractory DLBCL. CAR-T cell therapy is associated with immune effector-associated neurotoxicity syndrome (ICANS). ICANS symptoms range from the mild—headache, fatigue, and reversible aphasia—to severe and life-threatening symptoms—seizures, cerebral oedema, and death. Patients with CNS involvement were excluded from the registration trials due to concerns about potential excess toxicity, owing to the primary site of the lesion, the disrupted BBB, and the heavy burden of pre-treatment with CNS side effects such as whole-brain radiotherapy [82]. CAR-T cells have been shown to cross the BBB and can be detected at therapeutic levels for over 6 months post-infusion [53]. A 2022 meta-analysis of the toxicity and efficacy of CAR-T cell therapy in 128 patients with relapsed or refractory primary (30) and secondary (98) CNS lymphoma showed no increase in neurotoxicity when compared to that which was observed in the registration studies for DLBCL [83]. In addition, there was encouraging efficacy in PCNSL with 56% achieving a complete remission and 37% remaining in remission at 6 months.

In patients with CNS disease treated with CAR-T cells, there is a distinct neurotoxicity compared with CRS and ICANS, termed tumour inflammation-associated neurotoxicity (TIAN). It results from localized, “on-target” tumour-specific inflammation, with its manifestations reflecting the specific neuroanatomical location of the tumour [84]. It may result in increased intracranial pressure due to peritumoral therapy-related inflammation. The differentiation between ICANS and TIAN can be difficult but is fundamental, since the treatment approaches are distinct [85,86]. There has been a formal grading system proposed, but it is yet to be widely adopted [84]. The EBMT Practice Harmonisation and Guidelines Committee has recently published recommendations for “non-ICANS” neurological complications following CAR-T cell therapy, which include a management approach for TIAN [87].

Glofitamab, a CD20xCD3 bispecific antibody, has been shown to cross the blood–brain barrier and has shown efficacy in four patients with R/R DLBCL with secondary CNS involvement [88].

Pembrolizumab and nivolumab, both anti-PD1 antibodies, have demonstrated impressive results in case reports or small retrospective studies of patients with R/R PCNSL, but there are no prospective studies available [89,90,91,92,93].

## 7. Discussion

PCNSL is a devastating disease with significant morbidity and mortality. Its impact is pronounced because it significantly compromises function, and although there have been advances in treatment, the blood–brain barrier compromises treatment penetration. Additionally, the tolerability of intensive treatment in patients with functional impairment is limited.

Our case demonstrates how a relatively young patient was unable to tolerate the toxicity of intensive therapy and was considered unsuitable for consolidative autologous stem cell transplant or whole-brain radiotherapy. There are a considerable number of patients, either due to age, co-morbidities, or toxicity from induction therapy, who are unsuitable to receive consolidative therapy following intensive induction therapy. Progression-free and overall survival are known to be compromised in those who do not receive both intensive induction and consolidative therapy.

The decision to recommend an autologous stem cell transplant requires considering the potential risk of morbidity and mortality for individual patients balanced with the potential benefit of this procedure. There are validated tools, like the Hematopoietic Cell Transplantation-specific Comorbidity Index (HCT-CI), which help stratify risk. Patients also require normal organ function to reduce the potential risks. Adequate functional status is also important (usually an ECOG of less than 2), or at least a reasonable chance of improvement of functional status with treatment. As our patient developed significant renal and respiratory impairment, he was no longer eligible for stem cell transplant as consolidative therapy. Before BTKi availability, this patient would not have had further treatment and would have been offered best supportive care moving forward.

This case demonstrates that acalabrutinib, the BTKi inhibitor, was able to be used with no toxicity, with the patient remaining in remission for 18 months on this monotherapy. This facilitated recovery from the toxicities associated with intensive induction therapy, with subsequent improvement of both physical and cognitive function. It is unknown whether this patient would have remained in remission without acalabrutinib, though given that the initial induction therapy was truncated and that no standard consolidation was given, it would have been very unlikely for this patient to have remained in remission without additional therapy. Furthermore, it is unclear whether it is now safe to stop the acalabrutinib without risk of disease relapse. There is no evidence to support undertaking delayed autologous stem cell transplant after 18 months despite the improvement in his functional status and organ function; however, there was further discussion in the institutional MDT meeting, which recommended ongoing BTKi therapy. The patient continues to be monitored closely. If he shows signs of relapse, he will first be considered for available clinical trials; in the absence of a suitable trial, he would likely be offered salvage therapy with HD-methotrexate and cytarabine and considered for CAR-T cell therapy, which has become available as second-line therapy in Australia. He may also be considered again for autologous stem cell transplant if he continues to have normal organ function and appropriate performance status. If these treatments were unsuccessful, the patient would be offered best supportive care for palliation, highlighting the ongoing clinical need for improving therapies in this setting.

The data presented in this review show that acalabrutinib and the other discussed BTKis have evidence for crossing the blood–brain barrier and have shown promise in the refractory and relapsed disease setting for use as monotherapy or in combination with other therapies to improve outcomes. Certainly, using BTKis in PCNSL, which is largely of the MCD subtype (confirmed by MYD88 and/or CD79B molecular testing), is a rational drug choice, given that the MCD subtype is highly sensitive to BTK inhibition, although some studies suggest that BTKi benefit may be agnostic to MCD variants.

Whether BTKi can be used for those patients unable to have standard consolidation either due to age, functional status, or due to toxicity from initial therapy, or whether it might be used for maintenance, is a tantalising question that should be answered by clinical trials. Certainly, use for the patient in our case report has been favourable without toxicity. If shown to be effective in the clinical trial setting, further research is required regarding the duration of therapy: whether fixed duration will be sufficient, or if continuous therapy is necessary. Furthermore, the importance of testing for the molecular variants of MYD88 and CD79B from CSF and brain biopsy tissue should be incorporated into the design of clinical trials using BTKi to assess if these markers can enable more personalised therapy for patients to be determined or whether therapeutic effect is independent of the presence of these variants. The use of these molecular markers via ctDNA may also be used to guide the duration of therapy or predict those about to relapse, where early intervention may be of benefit.

## 8. Conclusions

There have been significant advances in the treatment of primary CNS lymphoma. However, there remains a need for additional therapies, particularly for those patients who are unsuitable for intensive chemotherapy. BTKis have a rational and promising role in this setting and may provide less toxic consolidation or maintenance therapies, either as monotherapies or in combination. Further research is needed.

## Figures and Tables

**Figure 1 ijms-26-10521-f001:**
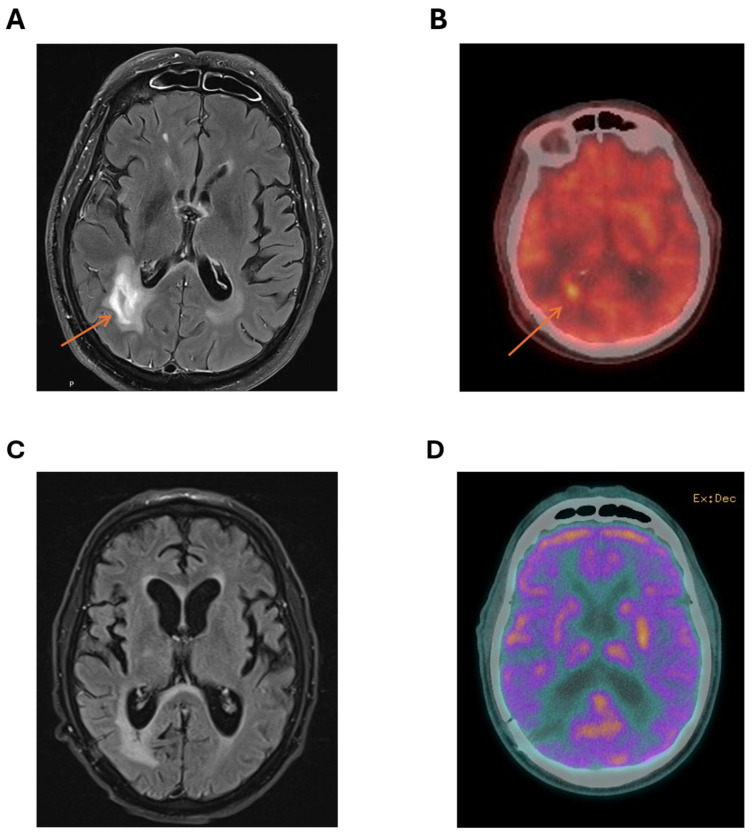
Characterisation of right parieto-temporal lesion at diagnosis with (**A**) MRI showing a T2 hyper-intense lesion (arrow pointing to lesion) and (**B**) PET scan with markedly increased standardized uptake value (SUV) in the area of the right parieto-temporal lesion (arrow pointing to lesion) compared to background FDG avidity. Imaging showing no residual-enhancing lesion after three cycles of MATRix with (**C**) MRI showing no T2 enhancement at the site of the original lesion and (**D**) cerebral PET showing no FDG-avid lesion in the right parieto-occipital area.

**Figure 2 ijms-26-10521-f002:**
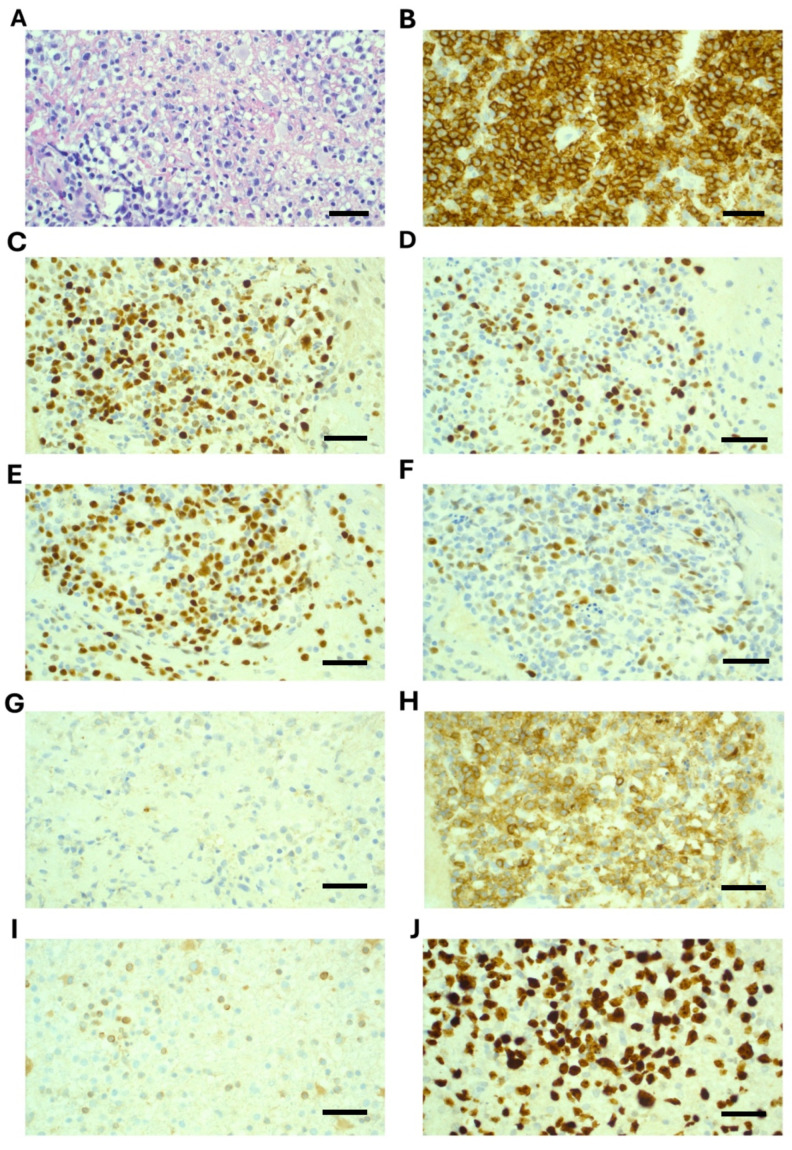
Histopathology and immunohistochemistry demonstrating diffuse large B-cell lymphoma from stereotactic biopsy of a right parietal brain lesion. (**A**) H&E, 40×, demonstrates disruption of brain tissue by large cells; (**B**) CD20, 40×, strongly positive; (**C**) BCL6, 40×, positive; (**D**) MUM1, 40×, positive in 40–50% of cells; (**E**) PAX5, 40×, positive; (**F**) c-Myc, 40×, 40–50% of cells are positive; (**G**) CD10, 40×, essentially negative (weak positive); (**H**) CD70, 40×, positive; (**I**) BCL2, 40×, negative; (**J**) Ki67, 40×, positive in 90% of cells. Scale bar is 50 um.

## Data Availability

The original contributions presented in this study are included in the article. Further inquiries can be directed to the corresponding author.

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
