# Peer review of "Acalabrutinib May Offer a New Therapeutic Approach for Consolidation and Maintenance of Primary CNS Lymphoma with Expression of MYD88 and CD79B Gene Variants: A Case Report and Literature Review of Primary CNS Lymphoma in the BTKi Era"

_ijms, 2025, doi:10.3390/ijms262110521_

Round 1
Reviewer 1 Report
Comments and Suggestions for Authors
Allison and colleagues report on a patient with PCNSL, who was successfully treated with acalabrutinib (2nd gen BTK Inh) after discontinuation of MATRix induction chemotherapy. Overall BTK inhibitors, particularly ibrutinib and tirabrutinib, have yoíelded encouraging efficacy in PCNSL. 2nd generation BTK inhibitor acalbrutinib is another interesting agent, which is currently being investigated. Please find my suggestions below:
- The case report is a bit lengthy and I would focus it on medically relevant information. Whether he was brought in by the police, found wandering the streets, etc adds little relevant information to the present article. Suggest to primarily focus the case report on PCNSL genetics, prior PCNSL treatment and subsequent acalabrutinib therapy.
- Could include a timeline as a panel in the imaging figure detailing the treatment course.
- As authors report on a novel targeted therapy I would strongly suggest to include and discuss important recent molecular studies on PCNSL (including: Radke J et al. Nat Comm 2022; Hernandez-Verdin I et al. Ann Oncol 2023; Hai L et al. Neuro Oncol 2025). Particularly the latter work has suggested that BCR and BTK signaling is particularly active in EBV negative CNS DLBCL whereas EBV positive cases rely on other pathways. This has important implications for the use of BTK inhibitors in PCNSL.
- Should add how molecular testing was performed and whether consent from the patient was obtained
- As the study focuses on PCNSL treated with a BTK Inh, I would remove details on SCNSL from the discussion.
- Similarly I would focus the discussion on novel targeted treatments in PCNSL rather than a summary of historical/conventional induction and consolidation treatments. Would probably also remove the paragraph on treatment in the elderly.
- When discussing CAR T therapy or immunotherapy would suggest to include TIAN as a relevant and distinct complication in patients with CNS lymphoma in addition to ICANS.
- When discussing non invasive diagnostics: please comment on different methodologies: e.g. detection of MYD88 mutations, CSF proteomics, methylation-based classification
Reviewer 2 Report
Comments and Suggestions for Authors
This is a good and well-established manuscript, offering a valuable case report and an informative literature review on Primary CNS Lymphoma in the BTKi era. The authors have clearly invested effort in presenting a relevant case and discussing the evolving therapeutic landscape.
However, based on the current presentation, I recommend a major revision for the following reasons:
- Introduction: While an "Introduction" section is indicated, it appears to segue directly into the case outline without a sufficiently developed and formal introductory paragraph that clearly sets the stage for the manuscript's objectives, the specific gap in knowledge it addresses, or the rationale behind the case report and literature review. A more robust introduction would significantly enhance the manuscript's foundation.
- The manuscript thoroughly discusses ASCT as the standard of care for fit patients in consolidation during the literature review. It also details the initial plan to consolidate the reported case patient with ASCT, and the subsequent reasons for not proceeding due to toxicities and comorbidities. However, beyond stating the general standard, the authors do not clearly articulate their overarching recommendation or specific criteria for when they would advise a patient to undergo consolidative ASCT versus alternative strategies (like the acalabrutinib used in their case). A clearer position, integrating the case's challenges with broader clinical decision-making, would be beneficial.
- The manuscript outlines salvage options for relapsed/refractory disease based on existing literature, including chemotherapy followed by consolidative ASCT. However, it does not explicitly detail the authors' proposed plan or strategy for patients who may relapse following the treatment approach highlighted in their case report or within their clinical practice. A more defined discussion on how they would manage subsequent relapses in the context of their proposed therapeutic avenues would strengthen the clinical utility of the manuscript.
Addressing these points will significantly improve the clarity, impact, and generalizability of the manuscript's findings and recommendations.
Round 2
Reviewer 1 Report
Comments and Suggestions for Authors
I would like to commend the authors for thoroughly revising their manuscript. Previous comments have been addressed. The review of literature is very comprehensive and I agree with the decision to keep the section on elderly patients.
Two minor comments prior to publication:
1) Please check that abbreviations are spelled out at first mention, e.g. 'VRF'.
2) May consider to include very recently published original research articles (PMID: 40663771) and a more recent, excellent review (PMID: 40179916) on TIAN/neurotox to make the review of literature even more comprehensive
Author Response
1) Please check that abbreviations are spelled out at first mention, e.g. 'VRF'.
Thank you. We have expanded abbreviations used in the text at first mention.
2) May consider to include very recently published original research articles (PMID: 40663771) and a more recent, excellent review (PMID: 40179916) on TIAN/neurotox to make the review of literature even more comprehensive.
Thank you. Great suggestions and very good new articles. We have made reference to them in our text.
Reviewer 2 Report
Comments and Suggestions for Authors
I accept as is.
Author Response
I accept as is.
Thank you for your help in reviewing our article. We appreciate your previous suggestions and are pleased you think it is now ready.